# Comparison of haemoglobin concentration measurements using HemoCue-301 and Sysmex XN-Series 1500: A survey among anaemic Gambian infants aged 6–12 months

**Mamadou Bah**[1,2]*, **Hans Verhoef**[2], **Abdou Camara**[1], **Morris Nden Ngom**[1], **Demba Jallow**[1], **Kebba Bajo**[1], **Foday Bah**[1], **Maarten Pleij**[2], **Maaike Klappe**[2], **Alasana Saidykhan**[1], **Emmanuel Okoh**[1], **Abdoulie Bah**[1], **Carla Cerami**[1]

1 Medical Research Council (MRC) Unit The Gambia at London School of Hygiene & Tropical Medicine (LSHTM), Fajara, The Gambia, 2 Division of Human Nutrition and Health, Wageningen University, Wageningen, The Netherlands

* mamadou.bah@lshtm.ac.uk

**Data Availability Statement:** Raw data are available in supplementary file 1 under supporting information

## Abstract

### Background

In low-income countries, point-of-care photometers are used in the screening and management of anaemia in individuals, but also in the assessment of population iron status when evaluating efficacy of intervention studies or public health interventions.

### Aims

We aimed to evaluate the accuracy of a commonly used photometer, HemoCue-301, in determining haemoglobin concentration among anaemic children aged 6–12 months in a field setting in rural Africa.

### Methods

This report concerns a secondary analysis of data from Gambian infants being screened for an ongoing randomized controlled trial. In those found to be anaemic by HemoCue-301, haemoglobin concentration was measured by Sysmex XN-1500, an automated haematology analyser that was used as a reference. Passing-Bablok regression analysis was used to estimate the regression constant (systematic deviation between two measurement methods that remain consistent across the range of measurements) and proportional bias (systematic deviation between two measurement methods that change in magnitude relative to the value being measured).

### Results

Analysis was based on 227 participants. There was strong evidence of absolute bias among moderately anaemic participants (haemoglobin concentration at 8.0g/dL) (absolute bias: 1.12g/dL; 95% CI: 0.91 to 1.37g/dL; proportional bias: 14.0%; 95% CI: 11.4% to 17.1%) in

**Funding:** This study is a secondary data analysis of baseline samples from a clinical trial funded by the Medical Research Council (grant reference: MR/R023360/1). CC, AMP, EO, AB, AC, MNG, DJ, KB, and AS are funded through MRCG@LSHTM. MB is partially funded through MRCG@LSHTM, Wageningen University, and MR/R023360/1. HV, MK, and MP are funded by Wageningen University. The funders had no role in study design, data collection and analysis, decision to publish, or preparation of the manuscript.

**Competing interests:** There is no competing interest to disclose.

haemoglobin concentrations measured by HemoCue-301 compared to those measured by Sysmex XN-Series1500. Bias was marginal at haemoglobin concentration of 11.0g/dL (absolute bias: -0.08g/dL; 95% CI: -0.18 to 0.07g/dL; proportional bias: -7.3%; 95% CI: -6.5% to 0.6%).

## Conclusion

Haemoglobin measurements by HemoCue-301 seem substantially biased in participants with haemoglobin less than 8.0g/dL.

## Introduction

According to WHO guidelines, paediatric blood transfusion should be performed on patients with haemoglobin less than 4.0 g/dl with clinical signs of severity [1]. However, few facilities in sub-Saharan countries have access to reliable haemoglobin tests with a turnaround time between 1 and 8 hours [2]. As a result, around 40–65% of requested transfusions are based on clinical judgment, such as severe pallor [2]. This clinical judgment is often inaccurate in identifying severe anaemia (haemoglobin less than 5.0 g/dl), leading to unnecessary transfusions [3, 4]. This does not only expose children to transfusion-related risks but may also contribute to blood scarcity for patients who are most in need [4, 5]. Therefore, the ability to provide rapid diagnosis of severe anaemia is critical for life-saving interventions and the prioritisation of blood units for those in most need. A study conducted in Kenya, Tanzania, and Uganda revealed that a delay in blood transfusion for severely anaemic children within the first 8 hours resulted in a 52% mortality rate, with 90% of these deaths occurring within the initial 2.5 hours [6]. This outcome was associated with insufficient blood units, emphasising the importance of managing and prioritising the already scarce blood resources to prevent avoidable deaths [6]. Conversely, in a Kenyan hospital, children receiving transfusion on the same day, compared to the following day, were associated with a 42% reduction in mortality [7]. Therefore, accurate measurement of haemoglobin concentrations is important in the screening and management of anaemia in individuals, but also in the assessment of population iron status in surveys to evaluate the efficacy of intervention studies or public health interventions. Automated haematology analysers are generally considered the reference standard. These analysers are expensive and require specific ambient temperatures, cold storage for reagents, and a stable source of electricity, which poses challenges in field settings [8]. To address these challenges, various battery-operated point-of-care photometers have been introduced. They rely on either pooled or single capillary blood samples, making them convenient for use in field settings and are widely used for country-wide anaemia prevalence studies [9].

One such photometer is the HemoCue-301 (HemoCue, Ängelholm, Sweden), which was specifically developed to withstand challenging environments and does not utilise any hygroscopic ingredients [10]. The HemoCue-301 utilises a double-wavelength method, measuring the absorbance of whole blood at 506nm the Hb/HbO2 isosbestic point (i.e., the wavelength where Hb and HbO2 have the same absorbance or intensity), and at a wavelength of 880nm to compensate for turbidity. However, this technique is sensitive to oxygen, resulting in a 1.3% increase in measured haemoglobin concentration per minute the sample is left in open air [11]. Therefore, readings must be completed within 20–30 seconds to ensure accuracy [10].

Despite their benefits, variability in haemoglobin measurements between these point-of-care devices and automated haematology analysers has been reported [2]. A systematic review

of 8 studies revealed higher haemoglobin concentration measurements of capillary blood using HemoCue-301 compared to measurement of venous blood in automated haematology analysers [11], with differences ranging from 0.5 to 6.1 g/L [11]. However, most of these studies determined the difference in haemoglobin measurement between the techniques using Bland-Altman analysis [12–20]. Unlike Bland-Altman analysis, Passing-Bablok regression analysis is a non-parametric method that does not assume constant measurement errors between methods and is robust against outliers [21].

We aimed to evaluate the accuracy of the HemoCue-301 in determining haemoglobin concentration among anaemic children aged 6–12 months in a field setting in rural Africa. As a reference, we used the Sysmex XN-1500 (Sysmex Corporation, Kobe, Japan), an automated haematology analyser that utilises a cyanide-free sodium lauryl sulphate reagent to lyse red blood cells, which, once released, forms a stable-coloured complex that is then measured photometrically [22].

## Materials and methods

### Ethical issues

This study contains an analysis of baseline data obtained from an ongoing efficacy trial of haem iron versus ferrous sulphate in anaemic (haemoglobin <11.0 g/dL) infants aged 6–12 months (see published protocol) [23]. Infants aged 6 to 12 months were identified in vaccination clinics and communities by trained field workers and individually informed about the study objectives and procedures. For illiterate participants, the study objectives and procedures were explained in the local language that the participants understood, in the presence of literate impartial witnesses and a written inform consent obtained.The trial received ethical approval from the Scientific Coordinating Committee of the Medical Research Council (MRC) Unit The Gambia, the Joint Gambia Government MRC Ethics Committee, and the Ethics Committee at the London School of Hygiene and Tropical Medicine (LEO27648) and adheres to the principles of Good Clinical Practice, with oversight from a Data Safety and Monitoring Board, and will be subject to monitoring by the Clinical Trials Office at the MRCG@LSHTM. The trial is registered with the Pan African Clinical Trial Registry (PACTR202210523178727) at www.pactr.org.

### Sample collection

Enrolment of participants started on the 11th of May 2023 in Jara West and Kiang East of The Gambian and concluded on the 1st of November 2023. After obtaining written informed consent from parents/guardians, capillary blood samples were collected in the field by a trained nurse. While seated, the participant's finger was cleaned, and non-fasting blood was collected by pricking the thumb with a lancet (Microtainer Contact-Activated Lancet, Becton-Dickinson, Franklin Lakes, NJ, USA). The initial capillary droplet was removed, and the second droplet was used to fill the microcuvette, ensuring that no air bubbles were introduced. Any excess blood on the sides of the microcuvette was wiped, and haemoglobin concentration was measured immediately by inserting the microcuvette into the HemoCue-301 photometer. Children with anaemia (haemoglobin <11.0g/dL by HemoCue-301) were invited for further screening and haemoglobin assessment was repeated using the reference method. For each participant, a 0.5mL venous blood sample was collected by a nurse into $K_2EDTA$ tubes (BD microtainer, reference number 365975), and samples were kept between 4˚C–8˚C until transferred to a central laboratory, where haemoglobin concentration was assessed within 4 hours of initial blood draw using a Sysmex XN-1500 analyser. Study data were collected and managed using RED-Cap electronic data capture tools hosted at MRC Unit The Gambia [24].

## Quality control

The HemoCue-301 is factory-calibrated against international reference samples and no further calibration is possible by the end user [25]. Weekly quality control checks were done for the HemoCue-301 utilising the control blood samples purchased for use with the Sysmex machine (XN Check, L1 catalogue number; 213484, L2 catalogue number; 213485, L3 catalogue number; 213486). No quality control failures were noted during the trial. The Sysmex XN-Series 1500 (Sysmex Corporation, Japan) instrument is housed in an accredited Good Clinical Laboratory Practice (GCLP) laboratory at MRCG at LSHTM at the Keneba field station. Daily quality checks and controls were conducted. All sample analyses were carried out by a trained scientific officer to ensure the accuracy and reliability of the data throughout the study.

## Statistical analysis

The statistical analysis was conducted in RStudio version 3.0. Passing-Bablok regression, a non-parametric method, was used to determine potential absolute and proportional bias using the Method Comparison Regression (mcr) package. Bias refers to non-random errors between measurements obtained from two different methods. Proportional bias refers to a systematic error in which the magnitude of the bias is directly proportional to the true value of the variable being measured. Unlike traditional linear regression, the Passing-Bablok regression method is insensitive to outliers and does not assume normal distribution, constant variance, or a linear relationship [21]. Passing-Bablok regression analysis was used to estimate absolute bias (arithmetic difference between two measurements that remains consistent across the range of measurements) and proportional bias (difference between two measurement methods relative to the value being measured). A difference of ±7% between methods as recommended by the College of American Pathologists and the Westgard Clinical Laboratory Improvement Amendments was considered acceptable [26].

According to the new WHO guidelines, normal as haemoglobin concentration >10.4 g/dL, mild anemia as hemoglobin concentrations between 9.5 and 10.4 g/dL, moderate anemia was defined as hemoglobin concentrations between 7.0 and 9.4 g/dL, and severe anemia as hemoglobin concentrations below 7.0 g/dL [27].

## Results

A total of 227 participants had their haemoglobin assessed using both the HemoCue-301 and Sysmex XN-Series 1500 and were included in the analysis (S1 File). The participants had a mean age of 7.5 months (SD: 1.6 months); 52% were female. Of 227 participants, 44% and 33% were mildly and moderately anaemic respectively, whereas, 22% were classified as nomal according to the Sysmex haematology analyser. One participant was classified as severely anaemic (haemoglobin <7.0 g/dl) according to the Sysmex haematology analyser but not the HemoCue-301.

The median haemoglobin concentrations for HemoCue-301 and Sysmex XN-Series were 10.2 g/dL (IQR: 9.8, 10.7 g/dL) and 9.8 g/dL (IQR:9.2, 10.4 g/dL), respectively. At haemoglobin concentration of 6.5 g/dL, Hemocue measurements showed an absolute bias of 1.72 g/dL (95% CI: 1.37 to 2.10 g/dL) and a proportional bias of 26.5% (95% CI: 21.1% to 32.3%). For haemoglobin concentration of 8.0 g/dL, the absolute bias was 1.12 g/dL (95% CI: 0.91 to 1.37 g/dL) and the proportional bias was 14.0% (95% CI: 11.4% to 17.1%), while haemoglobin concentration of 10.0 g/dL showed an absolute bias of 0.32 g/dL (95% CI: 0.26 to 0.42 g/dL) and a proportional bias of 3.2% (95% CI: 2.6% to 4.2%). No bias was found at haemoglobin concentration of 11.0 g/dL (absolute bias: -0.08 g/dL; 95% CI: -0.18 to 0.07; proportional bias:

-7.3%; 95% CI: -6.5 to 0.6) (Table 1, Fig 1). A modest positive correlation coefficient of 0.31, as indicated by Kendall's tau, indicates limited agreement between the measurement methods.

## Discussion

We demonstrated that haemoglobin measurements between HemoCue-301 and the Sysmex XN-1500 vary across different haemoglobin concentrations and the magnitude of bias is inversely proportional to haemoglobin concentration. There was strong evidence of a absolute bias in haemoglobin concentrations above the acceptable difference of ±7% set by the College of American Pathologists and Westgard Clinical Laboratory Improvement Amendments in moderate and severly anaemic participants [26].

The diagnostic accuracy of HemoCue-301 compared to automated hematology analyzers has been examined in several studies. While some studies have reported acceptable differences within ±7%, others have shown higher difference [11, 28]. We found that bias between methods was inversely associated with hemoglobin concentration. Similarly, a study conducted in Laotian children reported a higher bias in children with lower haemoglobin concentrations [29]. In blood donors, no bias was observed at a haemoglobin concentration of 12.0 g/dl when compared to the WHO reference standard haemoglobin-cyanide technique [17]. A systematic review of 8 studies among adults indicated higher HemoCue-301 haemoglobin concentration measurements compared to automated analysers. This study concluded that the difference fell within the acceptable limit of ±7% set by the College of American Pathologists and Westgard Clinical Laboratory Improvement Amendments [11]. However, it is important to acknowledge that differences in haemoglobin measurements vary between methods at different concentrations, as demonstrated by our study, which was not accounted for in this study.The Hemo-Cue-301 operates on spectrophotometric principles, where the absorbance of light passing through a blood sample is directly proportional to the concentration of haemoglobin at the Hb/HbO2 isosbestic point. Due to the stringent adherence to the haemoglobin measurement protocol within our clinical trial, the observed higher haemoglobin concentrations, particularly in anaemic children, are likely attributable to device bias arising from increased absorbance at lower haemoglobin concentrations rather than preanalytical errors. In contrast, a study among women in Peru has indicated lower haemoglobin concentrations measured by HemoCue-301 compared to automated analysers, with 50% of these women classified as anaemic according to HemoCue-301 [30]. However, in our analysis, only 7.4% of participants were misclassified as anaemic by the HemoCue-301, while they were classified as normal (haemoglobin concentrations greater than 11 g/dL) according to the Sysmex XN-1500.

The WHO is currently involved in ongoing research to comprehensively understand preanalytical and analytical factors influencing haemoglobin measurement using these point-of-care devices. This research aims to inform the development of new guidelines in this area [31] and our study will contribute to the existing evidence that can be utilised in these guidelines' development. This study is part of an ongoing clinical trial, thus ensuring stringent monitoring

**Table 1. Presents the absolute and proportional bias with their corresponding 95% confidence intervals, for haemoglobin concentration measurements obtained by HemoCue-301.** The absolute bias is expressed in grams per decilitre (g/dL), and the proportional bias is presented as a percentage (%).

| Sysmex Haemoglobin; g/dL | Absolute bias; g/dL (95% CI) | Proportional Bias; % (95% CI) |
|---|---|---|
| 6.5 | 1.72 (1.37 to 2.10) | 26.5 (21.1 to 32.3) |
| 8.0 | 1.12 (0.91 to 1.37) | 14.0 (11.4 to 17.1) |
| 10.0 | 0.32 (0.26 to 0.42) | 3.2 (2.6 to 4.2) |
| 11.0 | -0.08 (-0.18 to 0.07) | -7.3 (-6.5 to 0.6) |

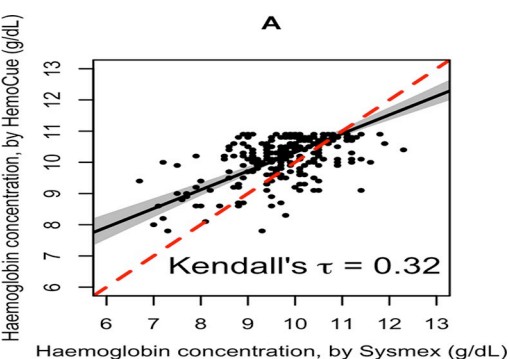

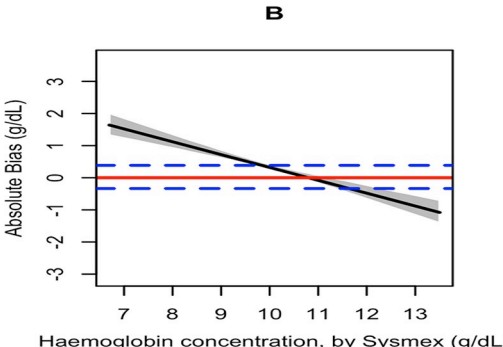

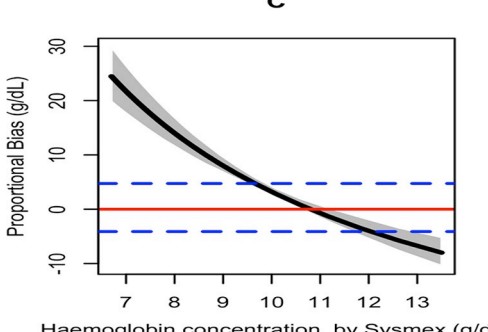

**Fig 1. Comparison of haemoglobin concentrations measured by HemoCue 301 vs Sysmex XN-1500.** HemoCue data were truncated at haemoglobin concentration <11.0g/dL (see text). **Panel A:**The black line represents the Passing-Bablok equation: $HaemoCue\ Haemoglobin = Intercept\ (4.32) + Slope(0.60) \times Sysmex\ Haemoglobin$; the red dashed line represents the identity line; the grey shade area indicates the 95% confidence band. **Panel B**: The black line represents the Passing-Bablok equation: $Absolute\ Bias = HemoCue\ method(Intercept + Slope \times Sysmex\ Haemoglobin) - (Sysmex\ Haemogloin(reference\ method))$; the red line signifies the zero bias, the blue dashed line represent ±7% acceptable difference and the grey shade indicates the 95% confidence interval. **Panel C:** The black line represents the Passing-Bablok equation: $Proportional\ bias(Xc) = \frac{Intercept+(Slope-1)\times Xc}{Xc}$. where the Intercept represents the constant bias at zero Sysmex (reference method), Slope reflects the proportional bias, and Xc denotes the specific Sysmex value for assessing percentage bias, the numerator combines the constant and proportional biases and divide it by Xc to normalizes the bias; The red dashed line signifies the zero bias, the blue dashed line represent ±7% acceptable difference, and the grey shade indicates the 95% confidence interval.

and adherence to trial protocols throughout the blood sampling process and haemoglobin measurements. This study has several limitations as it was a secondary data analysis using a convenience sampling method. Consequently, the findings may not be generalisable to rural Gambian children aged between 6 and 12 months. Additionally, capillary and venous blood sampling for haemoglobin measurement were not performed on the same day. The retrospective nature of our study limited our ability to explore the potential impact of different blood sample types (venous vs capillary) on haemoglobin concentration measurements. Furthermore, our study design did not allow for a direct comparison between point-of-care tests and automated haematology analysers regarding anaemia prevalence. This is because only children identified as anaemic by HemoCue-301 underwent haemoglobin measurements using the automated analyser. Another limitation is the comparison of different blood types using samples collected on different days.

## Conclusion

Our study indicated that bias due to HemoCue-301 depends on the actual Hb concentration, and seems unacceptably high at Hb values below 8.0 g/dL. While this method may offer convenience in field surveys for estimating anaemia prevalence, caution is required when using it for a screen-and-treat approach.

## Supporting information

**S1 File. Raw dataset used in this analysis.**
(PDF)

## Acknowledgments

We thank the IDeA3 study team at MRCG@LSHTM for their dedicated support, the MRCG Keneba laboratory, the data management team and the MRCG Clinical Trial Unit. We also thank the participants and communities of Kiang East and Jarra West.

## Author Contributions

**Conceptualization:** Mamadou Bah, Carla Cerami.

**Data curation:** Mamadou Bah, Carla Cerami.

**Formal analysis:** Mamadou Bah, Hans Verhoef, Carla Cerami.

**Funding acquisition:** Carla Cerami.

**Investigation:** Mamadou Bah, Hans Verhoef, Carla Cerami.

**Methodology:** Mamadou Bah, Hans Verhoef, Carla Cerami.

**Project administration:** Mamadou Bah.

**Supervision:** Hans Verhoef, Carla Cerami.

**Validation:** Mamadou Bah, Carla Cerami.

**Visualization:** Mamadou Bah, Hans Verhoef, Carla Cerami.

**Writing – original draft:** Mamadou Bah, Hans Verhoef, Demba Jallow, Carla Cerami.

**Writing – review & editing:** Mamadou Bah, Hans Verhoef, Abdou Camara, Morris Nden Ngom, Demba Jallow, Kebba Bajo, Foday Bah, Maarten Pleij, Maaike Klappe, Alasana Saidykhan, Emmanuel Okoh, Abdoulie Bah, Carla Cerami.

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
