## [Decision Letter · Decision Letter 0]

16 Sep 2024

PONE-D-24-33800Comparison of haemoglobin concentration measurements using HemoCue-301 and Sysmex XN-Series 1500: a survey among anaemic Gambian infants aged 6-12 monthsPLOS ONE

Dear Dr. Bah,

Thank you for submitting your manuscript to PLOS ONE. After careful consideration, we feel that it has merit but does not fully meet PLOS ONE’s publication criteria as it currently stands. Therefore, we invite you to submit a revised version of the manuscript that addresses the points raised during the review process.

We look forward to receiving your revised manuscript.

Kind regards,

Benedikt Ley, PhD

Academic Editor

PLOS ONE

Journal Requirements:

4. Thank you for stating the following in your manuscript: 

“This study was funded by a grant from the UK Research & Innovation (UKRI) to the Medical Research Council Unit The Gambia at London School of Hygiene & Tropical Medicine (grant reference: MR/R023360/1). CC, AMP, EO, AB, AC, MNG, DJ, KB and AS are funded through MRCG@LSHTM. MB is partially funded through MRCG@LSHTM, Wageningen University and MR/R023360/1. HV, MK, and MP are funded by Wageningen University.

Role of the funding source

The funder had no role in the design or implementation of the study.”

5. In the online submission form, you indicated that “Data are available from the PI upon reasonable request”.

1) In a public repository, 

2) Within the manuscript itself, or 

3) Uploaded as supplementary information.

6. Please ensure that you refer to Figure 1 in your text as, if accepted, production will need this reference to link the reader to the figure.

7. Please remove your figures from within your manuscript file, leaving only the individual TIFF/EPS image files, uploaded separately. These will be automatically included in the reviewers’ PDF.

8. We note you have included a table to which you do not refer in the text of your manuscript. Please ensure that you refer to Table 1 in your text; if accepted, production will need this reference to link the reader to the Table.

**Additional Editor Comments:**

Please extend your limitations section substantially as suggested by reviewer 2. Please include limitations around comparing different sources of blood and the fact that blood was collected on different days. 

Reviewers' comments:

Reviewer's Responses to Questions

**Comments to the Author**

1. Is the manuscript technically sound, and do the data support the conclusions?

Reviewer #1: No

Reviewer #2: Yes

2. Has the statistical analysis been performed appropriately and rigorously? 

Reviewer #1: No

Reviewer #2: I Don't Know

3. Have the authors made all data underlying the findings in their manuscript fully available?

Reviewer #1: Yes

Reviewer #2: Yes

4. Is the manuscript presented in an intelligible fashion and written in standard English?

Reviewer #1: Yes

Reviewer #2: Yes

5. Review Comments to the Author

Reviewer #1: • Methods

o Please include definitions (Hb level ranges) of severe, moderate, mild anaemia

o Kindly describe the definitions of Absolute Bias and Proportional Bias used in this study (the equations in the caption of Figure 1) in this section, so the readers know where the number in the Results section came from

o Lines 132-133: “We defined absolute and proportional bias when the confidence intervals excluded 0 and 1 for the intercept or the slope respectively.” I am a bit unsure of what this sentence means; perhaps reword it to clarify?

• Results

o Table 1 and Figure 1 were not referred to in the text

o Participant demography: how many participants have severe, moderate, mild, and no anaemia? Also consider describing age, sex, and other factors that may impact Hb levels among participants

o Lines 143-144: “One participant was classified as severely anaemic (haemoglobin <7.0 g/dl) …” In line 149 severe anaemia was defined as 6.5 g/dL, this is contradictory

o Lines 148-151: “For severe anaemia (haemoglobin concentration at 6.5 g/dL), Hemocue measurements showed an absolute bias of 1.72 g/dL (95% CI: 1.37 to 2.10 g/dL) and a proportional bias of 26.5% (95% CI: 21.1% to 32.3%).” Is the listed absolute and proportional bias applicable to 6.5 g/dL or to the entire range of severe anaemia (anything below 6.5 G/dL)?

o Lines 162-163: “the magnitude of bias is inversely proportional to haemoglobin concentration” Based on Figure 1, the bias was also increasing but to the negatives the higher the Hb concentration

o Lines 170-174: This is a repeat of what has been written in the Results section, consider removing

o Lines 174-175: “… while those with a haemoglobin concentration of 11.0 g/dL showed marginal or non-existent bias.” What made you say this? What was the limit of being "marginal"? At 11 g/dL according to Table 1, the proportional bias was -7.3%, which was outside of the acceptable limit of ±7.0%.

o Lines 175-176: “Similarly, a study conducted in Laotian children reported a higher bias in anaemic compared to non-anaemic children (22).” Only anaemic children were enrolled for this study, how is this comparable to the Laotian study?

o Lines 199-215: Consider putting these in the Introduction/Background section instead

• Conclusion

o Lines 242-243: “Our study indicated that bias due to HemoCue-301 depends on the actual Hb concentration, and seems unacceptably high at Hb values below 8.0 g/dL.” At 8.0 g/dL the proportional bias was 14.0%, which was also beyond the acceptable value of ±7.0%. Consider calculating a range of Hb levels where the proportional bias is acceptable (±7.0%) instead.

Reviewer #2: This manuscript compared POC Hb measurement HemoCue 301 and Sysmex XN-1500 as reference. The authors compared hemoglobin measurement from capillary blood using HemoCue 301 and those children who are anaemic were re-measured using venous blood and Sysmex XN-1500. There are limitations to the study that the authors have admitted, namely, the measurement of capillary and venous blood was not done on the same day as an example. I would also like to point out that it is known capillary blood to have higher concentration of hemoglobin dan venous blood and this was actually seen in this study. I think the authors should have compared measurement of venous blood using HemoCue 301 AND Sysmex XN-1500 rather than using different blood source and different devices.

Capillary blood sometimes give inconsistent results depending how you get the capillary blood. If only small amount of blood can be drawn out (in case of anemia), you tend to press the fingers more to get the enough blood into the cassette which will result in a more diluted blood since more plasma is squeezed, not hemoglobin.

So I think, the authors should also put these into their limitations of the study.

6. PLOS authors have the option to publish the peer review history of their article (what does this mean?). If published, this will include your full peer review and any attached files.

Reviewer #1: No

Reviewer #2: No

---

## [Author Response · Author response to Decision Letter 0]

29 Sep 2024

Review of the manuscript: (Line numbers refer to version without track changes)

"Comparison of haemoglobin concentration measurements using HemoCue-301 and Sysmex XN-Series 1500: a survey among anaemic Gambian infants aged 6-12 months" (PNTD-D-24-33800)

Response to Editors comments

Response: Style requirements have been updated.

Response: Additional details about consent is added in lines 101–106.

3. Thank you for stating the following in your manuscript: “This study was funded by a grant from the UK Research & Innovation (UKRI) to the Medical Research Council Unit The Gambia at London School of Hygiene & Tropical Medicine (grant reference: MR/R023360/1). CC, AMP, EO, AB, AC, MNG, DJ, KB and AS are funded through MRCG@LSHTM. MB is partially funded through MRCG@LSHTM, Wageningen University and MR/R023360/1. HV, MK, and MP are funded by Wageningen University.

Role of the funding source:

The funder had no role in the design or implementation of the study.”

Response: This is now added in the cover letter: This study is a secondary data analysis of baseline samples from a clinical trial funded by the Medical Research Council (grant reference: MR/R023360/1). CC, AMP, EO, AB, AC, MNG, DJ, KB, and AS are funded through MRCG@LSHTM. MB is partially funded through MRCG@LSHTM, Wageningen University, and MR/R023360/1. HV, MK, and MP are funded by Wageningen University.

Funding-related text is removed from the manuscript.

4. In the online submission form, you indicated that “Data are available from the PI upon reasonable request.”

All PLOS journals now require all data underlying the findings described in their manuscript to be freely available to other researchers, either:

2. Within the manuscript itself, or Uploaded as supplementary information.

Response: Raw data is available in supplementary file 1

5. Please remove your figures from within your manuscript file, leaving only the individual TIFF/EPS image files, uploaded separately. These will be automatically included in the reviewers’ PDF.

Response: Figures are removed from the manuscript file.

6. We note you have included a table to which you do not refer in the text of your manuscript. Please ensure that you refer to Table 1 in your text; if accepted, production will need this reference to link the reader to the Table.

Response: Table 1 is now referred to in line 178.

7. Please extend your limitations section substantially as suggested by reviewer 2. Please include limitations around comparing different sources of blood and the fact that blood was collected on different days.

Response: This has been added in lines 253–254.

Reviewer 1 Comments

1. Methods

a. Please include definitions (Hb level ranges) of severe, moderate, mild anaemia.

Response: Added to lines 157–160.

b. Kindly describe the definitions of Absolute Bias and Proportional Bias used in this study (the equations in the caption of Figure 1) in this section, so the readers know where the numbers in the Results section come from.

Response: Definitions added in lines 151–154.

c. Lines 132–133: “We defined absolute and proportional bias when the confidence intervals excluded 0 and 1 for the intercept or the slope respectively.” I am unsure what this sentence means; perhaps reword it to clarify?

Response: Clarified in lines 151–154.

2. Results

a. Table 1 and Figure 1 were not referred to in the text.

Response: Thank you for pointing this out. We have now referred to these in line 178.

b. Participant demography: how many participants have severe, moderate, mild, and no anaemia? Also, consider describing age, sex, and other factors that may impact Hb levels among participants.

Response: This is now described in 162 to 168.

c. Lines 143–144: “One participant was classified as severely anaemic (haemoglobin <7.0 g/dl) …” In line 149, severe anaemia was defined as 6.5 g/dL. This is contradictory.

Response: The results section has been reworded for clarity.

d. Lines 148–151: “For severe anaemia (haemoglobin concentration at 6.5 g/dL), Hemocue measurements showed an absolute bias of 1.72 g/dL (95% CI: 1.37 to 2.10 g/dL) and a proportional bias of 26.5% (95% CI: 21.1% to 32.3%).” Is the listed absolute and proportional bias applicable to 6.5 g/dL or the entire range of severe anaemia (anything below 6.5 g/dL)?

Response: This applies only to haemoglobin concentrations of 6.5 g/dL. The results section has been reworded.

e. Lines 162–163: “The magnitude of bias is inversely proportional to haemoglobin concentration.” Based on Figure 1, the bias was also increasing but to the negatives at higher Hb concentrations.

Response: Yes, you are correct. The bias is inversely related to haemoglobin concentration. For example, at lower haemoglobin concentration, the bias tends to be higher, but it increases to the negative end at higher haemoglobin concentration. The potential reason is described in line 225 to 231.

f. Lines 170–174: This is a repeat of what has been written in the Results section; consider removing.

Response: Lines 170–174 have been removed as suggested.

g. Lines 174–175: “… while those with a haemoglobin concentration of 11.0 g/dL showed marginal or non-existent bias.” What made you say this? What was the limit of being "marginal"? At 11 g/dL, according to Table 1, the proportional bias was -7.3%, which was outside the acceptable limit of ±7.0%.

Response: This is because the lower limit is outside the acceptable limit of -7.0% whereas, the upper limit is within the +7.0%.

h. Lines 175–176: “Similarly, a study conducted in Laotian children reported a higher bias in anaemic compared to non-anaemic children (22).” Only anaemic children were enrolled in this study. How is this comparable to the Laotian study?

Response: Thank you for this comment. We have added in additional information about the Laotian study and its relevance in lines 215–217.

i. Lines 199–215: Consider putting this in the Introduction/Background section instead.

Response: Moved to the introduction section (line 48–64) as suggested.

3. Conclusion

a. Lines 242–243: “Our study indicated that bias due to HemoCue-301 depends on the actual Hb concentration and seems unacceptably high at Hb values below 8.0 g/dL.” At 8.0 g/dL, the proportional bias was 14.0%, which was also beyond the acceptable value of ±7.0%. Consider calculating a range of Hb levels where the proportional bias is acceptable (±7.0%) instead.

Response: Fig 1B and C shows the different ranges of haemoglobin concentrations and corresponding biases, and Table 1 shows biases at a given Sysmex haemoglobin concentration.

Reviewer 2 Comments

1. Capillary blood sometimes gives inconsistent results depending on how it is collected. In cases of anaemia, if only a small amount of blood can be drawn, you may press the fingers more to get enough blood into the cassette, which results in a more diluted sample since more plasma is squeezed out.

Response: This limitation has been added in lines 253–254.

---

## [Decision Letter · Decision Letter 1]

21 Oct 2024

Comparison of haemoglobin concentration measurements using HemoCue-301 and Sysmex XN-Series 1500: a survey among anaemic Gambian infants aged 6-12 months

PONE-D-24-33800R1

Dear Dr. Bah,

We’re pleased to inform you that your manuscript has been judged scientifically suitable for publication and will be formally accepted for publication once it meets all outstanding technical requirements.

Kind regards,

Benedikt Ley, PhD

Academic Editor

PLOS ONE

Additional Editor Comments (optional):

Reviewers' comments:

Reviewer's Responses to Questions

**Comments to the Author**

1. If the authors have adequately addressed your comments raised in a previous round of review and you feel that this manuscript is now acceptable for publication, you may indicate that here to bypass the “Comments to the Author” section, enter your conflict of interest statement in the “Confidential to Editor” section, and submit your "Accept" recommendation.

Reviewer #1: All comments have been addressed

Reviewer #2: All comments have been addressed

2. Is the manuscript technically sound, and do the data support the conclusions?

Reviewer #1: Yes

Reviewer #2: Yes

3. Has the statistical analysis been performed appropriately and rigorously? 

Reviewer #1: Yes

Reviewer #2: Yes

4. Have the authors made all data underlying the findings in their manuscript fully available?

Reviewer #1: (No Response)

Reviewer #2: Yes

5. Is the manuscript presented in an intelligible fashion and written in standard English?

Reviewer #1: Yes

Reviewer #2: Yes

6. Review Comments to the Author

Reviewer #1: Summary

This article evaluated the accuracy of the point-of-care Hb measurement assay HemoCue-301 in measuring Hb levels in anaemic children against the reference assay Sysmex XN-1500. The absolute and proportional differences between measurements from the two assays were analysed with the Passing-Bablok regression. This article found that larger bias/differences were present in lower Hb levels, that were outside of the acceptable difference of ±7.0%.

Comments

All of my previous comments have been addressed satisfactorily.

Reviewer #2: Thank you for addressing the comments. Evaluating Hemocue 301 vs gold standard Sysmex is important because right now, especially in LMIC, are using Hemocue instead of blood analyzer for Hb measurement. Knowing that there is a strong bias for Hb below 8 g/dL is important finding.

I would still prefer the authors to compare venous blood using Sysmex and Hemocue however, as it is, we can already see the result.

7. PLOS authors have the option to publish the peer review history of their article (what does this mean?). If published, this will include your full peer review and any attached files.

Reviewer #1: No

Reviewer #2: No

---

## [Editor Report · Acceptance letter]

26 Oct 2024

PONE-D-24-33800R1 

PLOS ONE

Dear Dr. Bah, 

I'm pleased to inform you that your manuscript has been deemed suitable for publication in PLOS ONE. Congratulations! Your manuscript is now being handed over to our production team.

Kind regards, 

on behalf of

Dr Benedikt Ley 

Academic Editor

PLOS ONE